# Effect of Short Fermentation Times with *Lactobacillus paracasei* in Rheological, Physical and Chemical Composition Parameters in Cassava Dough and Biscuits

**Samuel Longoria** [1] , **Juan Contreras** [2], **Ruth Belmares** [2],*  , **Mario Cruz** [3],*  **and Mildred Flores** [3]

1   Departamento de Nutrición, Vicerrectoría de Ciencias de la Salud, Universidad de Monterrey, Avenida Morones Prieto 4500 Pte, C.P., San Pedro Garza García, Nuevo León 66238, Mexico; samuel.longoria@udem.edu

2   Departamento de Investigación en Alimentos, Facultad de Ciencias Químicas, Universidad Autónoma de Coahuila Blvd. V. Carranza e Ing. José Cárdenas sin numero, Col. República, C.P., Saltillo, Coahuila 25280, Mexico; carlos.contreras@uadec.edu.mx

3   Departamento de Ciencia y Tecnología de Alimentos, Universidad Autónoma Agraria Antonio Narro, Calzada Antonio Narro 1923, Col. Buena Ventura, C.P., Buenaventura, Coahuila 25315, Mexico; labcta@hotmail.com

*   Correspondence: ruthbelmares@uadec.edu.mx (R.B.); myke13_80@hotmail.com (M.C.)

**Abstract:** Dough fermentation with lactic acid bacteria has been extensively studied due to the associated health benefits and its effects on physical and rheology parameters in dough and bread. However, most of the studies rely on long fermentation times. The aim of this study is to evaluate the effect of short fermentation times (0 to 8 h) with *Lactobacillus paracasei* in rheology, physical and chemical properties on cassava dough and biscuits. Both storage modulus and loss modulus decreased as the fermentation times increased, down to 54,206.67 ± 13,348 and 17,453.89 ± 3691 Pa, respectively. Fermentation with *L. paracasei* influenced biscuit's hardness and chemical properties, and gas cell sizes were increased notably. These results suggest that short fermentation times could be used to improve dough's rheological characteristics.

**Keywords:** gluten-free; cassava; fermentation; storage modulus; loss modulus; *Lactobacillus paracasei*

## 1. Introduction

Bread and biscuits are among the most popular foodstuffs worldwide [1]. Gluten is the protein complex involved in the development of bread's volume and springiness. Structurally, gluten provide the network in which starch will gelatinize during baking, leading to bread's most common physical characteristics [2]. However, a significant amount of the worldwide population is not able to consume products containing this protein complex.

Celiac disease is an autoimmune disorder which shows gluten intolerance. Characteristic symptoms are chronic inflammation and atrophy of intestinal villi, impairing digestion and absorption processes [3]. Therefore, there is an increasing demand for gluten free (GF) products due to the attention celiac disease is receiving worldwide. There are several approaches researchers have taken to develop GF products, they vary from complex ones like using proteases to lower gluten content [4] to simpler ones such as using GF flours from alternative sources such as rice, chestnut and cassava [5–7].

Cassava (*Manihot esculenta*), which has been used in several foodstuffs including bread, is one of the most economical sources of carbohydrates and energy and its production is increasing [8].

Due to its high carbohydrate content, it could be used potentially as a sourdough substrate to improve physical and nutritional characteristics of products made from cassava.

Sourdough fermentation with lactic acid bacteria (LAB) has been reported to provide several health enhancing components such as γ-aminobutyric acid [9] and to modify rheological and physical characteristics of GF products [10]. However, one of sourdough's downsize presently is the high amount of time consumed during fermentation.

Several LAB have been employed in the development of sourdough, most commonly *Lactobacillus plantarum*. However, *L. paracasei* has seen increasing interest due to its health benefits on colon cancer [11] and for its use as sourdough starter [12].

The objective of this work is to evaluate the effect, in rheology, physical and chemical properties, on dough and biscuits fermented with *L. paracasei* for short fermentation times, in order to assess a faster method of enhancing biscuit's physical characteristics.

## 2. Materials and Methods

Fresh cassava tubers and commercial calcium enriched agave's inulin (enature®) were bought in a local supermarket in Saltillo, Coahuila, Mexico.

### 2.1. Flour Production

Cassava tubers were washed, peeled and chopped into pieces. The pieces were dried in an air oven at 80 °C during 48 h. Afterwards, they were milled using a Pulvex Mini100 mill and screened into a maximum particle size of 0.42 mm. Cassava flour was stored in sealed glass bottles at room temperature until use.

### 2.2. Dough Fermentation

Dough fermentation was carried out using *L. paracasei* obtained through local supplier Abiasa México S.A. de C.V. in a $1 \times 10^8$ CFU/g of flour concentration.

Probiotics were reactivated in water at 37 °C for 5 min following manufacturer's instructions. Five different fermentation times were selected, (0, 2, 4, 6 and 8 h). The formulation used contains (all per 100 g cassava flour): 9.6 g of whole egg, 1.1 g of sodium bicarbonate, 0.3 g of salt, 84 g of water and 50 g of an animal fat source (34 g fats, 35 g water, 28 g protein, 3.5 g carbohydrates, all per 100 g of animal fat source).

pH was measured to assure the development of probiotics in the dough according to the Mexican normative NMX-F-317-S-1978 [13].

Doughs were fermented in an GI2-2 Shel Lab General purpose incubator at 37 °C.

### 2.3. Dough and Biscuit Preparation

After fermentation and homogenization, dough balls were shaped and weighted into three equal parts. Biscuits were baked at 180 °C for 20 min in a San Son HCX Plus3 convection oven. After baking, biscuits were left to cool at room temperature for 30 min. After cooling, color, hardness and moisture tests were performed. Biscuits were then dried at 35 °C in an air oven for 24 h, grinded and stored in light protected sealed glass bottles at room temperature until use.

### 2.4. Rheological Assays

Doughs were analyzed in a Physica MCR 501 oscillatory rheometer according to the procedure described [14], with some modifications using a parallel plate geometry at 25 °C with a probe diameter of 25 mm and gap of 0.5 mm. Readings were taken for storage modulus (G′) and loss modulus (G′′) in the linear viscoelastic region during an angular frequency sweep (ω) ranging from 0–100 1/s. All assays were conducted by triplicate. The damping factor (tan δ) was calculated using the following formula:

$$\tan \delta = \frac{G''}{G'}$$

The shear stress (τ) was calculated using the following formula:

$$\tau = \frac{3T}{2\pi R^3}$$

where T = torque and R = probe's radium.

### 2.5. Physical Assays

Biscuit's color values were obtained by using a 3nh NR2OXE Precision colorimeter. The scanning was performed at the center of the biscuit by triplicate, obtaining the mean of lightness (L*), redness (a*) and yellowness (b*) values. Hardness was determined by using an EXTECH FHT200 Penetrometer on freshly baked biscuits using a probe of 3 mm. The test was conducted at the center of the biscuits obtaining the mean in newtons (N). All tests were conducted by triplicate.

### 2.6. Chemical Composition Evaluation

All assays were conducted by triplicates. Crude fiber, fat, ash and protein content were determined by A.O.A.C. 14.111, 14.019, 14.103 and 14.108 methods, while carbohydrate content was obtained using FAO's differential method [15]. Moisture was determined by using an MB23 Ohaus thermobalance; conditions were set to 110 °C during 10 min with 0.5 g of sample.

### 2.7. Experimental Design and Statistical Analyzes

A completely randomized design was applied to evaluate all parameters along with the different fermentation times. An analysis of variance (ANOVA) was performed along with Tukey's tests to establish statistically significant differences between means of all variables. Analyses were performed in InfoStat ver. 2015e statistical software with a significance level (α) of 0.05.

## 3. Results and Discussion

### 3.1. Dough Fermentation

In Figure 1, pH results of the fermented doughs are presented. The main objective of this study was to use reduced fermentation times for easier scaling up to industrial applications. pH was measured every 2 h from 0 to 6 with a final measurement at 24 h to evaluate *Lactobacillus paracasei* fermentation.

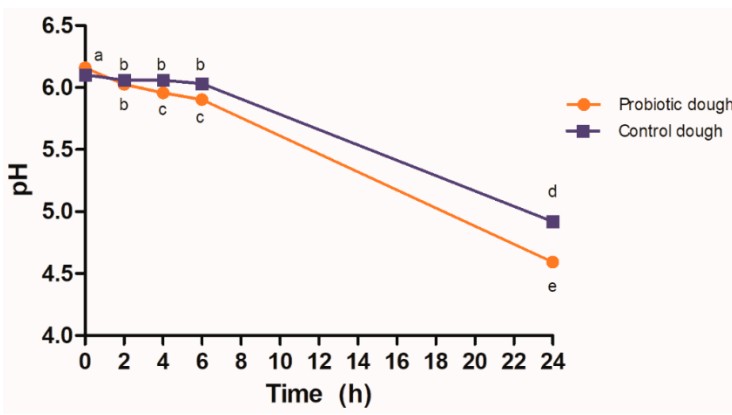

**Figure 1.** pH values for dough fermentation; data is mean (±standard deviation) of three replicates. a, b, c, d, e different letters indicate significant differences ($p < 0.05$).

Statistically significant differences between control dough and fermentation dough can be seen parting from 4 h of fermentation, being 6.06 ± 0.00 and 5.95 ± 0.005. Reducing processing times, while providing health benefits, could be of interest in the food industry. Other authors have reported lower pH values obtained after a 24 h fermentation, these values being 2.90 and 3.5, respectively [16,17]; while in this study, the pH values obtained were 4.59 ± 0.005 and 4.92 ± 0.00, respectively, for fermented and control dough. Direct comparisons are difficult, because the elements employed, such as flour source, probiotic microorganisms and its concentration, are different. Generally speaking, a lower pH value is desirable since it is used as an indirect indicator of microorganism proliferation and thus higher biotransformation of health beneficial compounds, as well as dough's rheology modification.

Control dough pH results suggest that natural cassava's microbiota could be fermenting carbohydrates present in the flour, however, not achieving the same results as the probiotic dough. Several lactic acid bacteria have been reported to be commonly found in cassava, such as *L. plantarum*, *L. amylolyticus* and *L. mannihotivorans* [18–20]. As far as we are concerned, there are no reports of an antagonistic effect between *L. paracasei* and the most common LAB in cassava. They have even been used together in other foodstuffs [21]. This suggests that pH differences can be attributed to *L. paracasei* inoculation used during this study.

*L. plantarum* has shown to be one of the lactic acid bacteria to adapt easily to cassava sourdough fermentation [22]. However, to the best of our knowledge, there is no study making a comparison in cassava sourdough between *L. plantarum* and *L. paracasei*.

## 3.2. Rheology Assays

$\overline{G}'$ provides information about the deformation energy stored in the sample. A higher value in this parameter means a higher elastic or solid-like behavior; while $\overline{G}''$, on the contrary, is the energy lost and the sample does not regain its original shape due to its viscous behavior [23].

One of the main concerns involving GF products are the issues associated with high carbohydrate concentrations as well as low protein ones, especially gluten. This could often lead to lower quality products due to poor development during proofing [24]. Lower G' values are desired when considering doughs for bread and similar products, while higher ones are desired for cookies and cookie-like products. These values could serve as a predictive indicator of the suitability of the dough to be used for a specific type of product.

In Figures 2 and 3 the results for the trend in storage and loss modulus are presented. In this work, $\overline{G}'$ was higher than $\overline{G}''$ in all fermentation times used. This falls in line with what has been seen in different gluten-free biscuits [14,25]. A decline can be observed in both $\overline{G}'$ and $\overline{G}''$ along the increase of fermentation time. One possible explanation is pH. Dough acidification impacts chemical components in the food matrix, allowing better interaction between water molecules and structure-forming components such as proteins and starch [14]. These results are similar to those found in a study using a traditional sourdough bread where an increase of moisture was found by acidifying the dough [19,26].

In both Figures 2 and 3, an increase in both G' and G'' respectively can be observed as the angular frequency increases. This behavior has been reported in another study [14] and attributed to increased dough structure. In both cases, fermented doughs have a noticeably different trend compared to control dough (0 h). Authors often explain this behavior by the amount of organic acids produced during fermentation [14].

Doughs present weak gel-like behavior according to the tan δ values shown in Figure 4 (6 h and 8 h), while 2 h, 4 h and 6 h are right in the boundary between a weak gel and an excess of structuring components [24]. Similarly, this could be caused by the acidification of the dough, allowing better interaction between water and other molecules present, as previously mentioned. The decline of τ indicates that dough opposes less resistance as the fermentation time increases, meaning that doughs start exhibiting a decreased elastic behavior compared to control. This has been discussed in

another study where it was found that sourdoughs have lower shear stress values compared to regular doughs [27].

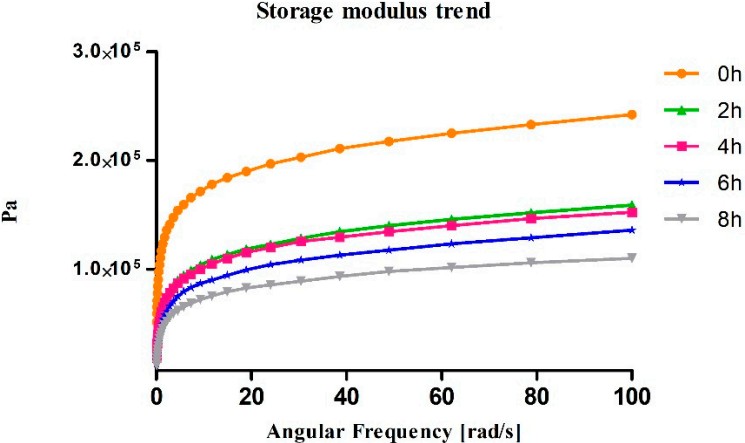

**Figure 2.** Storage modulus trends obtained through the rheology assay in fermented doughs.

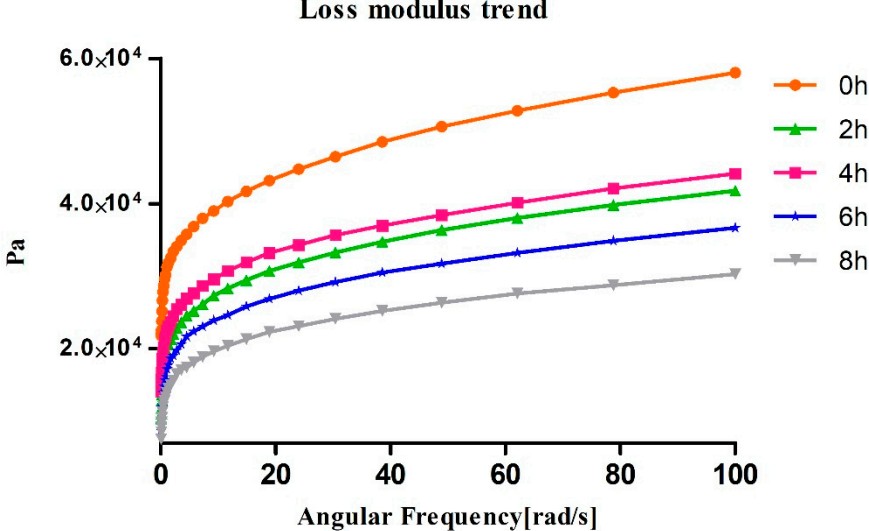

**Figure 3.** Loss modulus trends obtained through the rheology assay in fermented doughs.

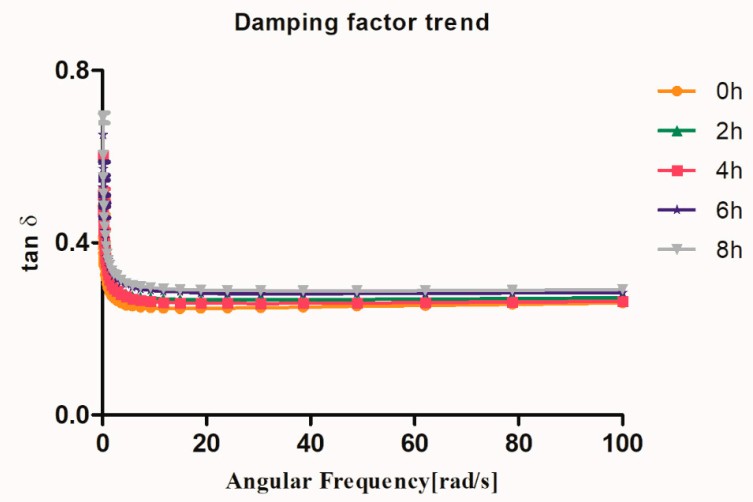

**Figure 4.** Damping factor trends obtained through the rheology assay in fermented doughs.

### 3.3. Physical Assay

Hardness values obtained from sourdough bread often show mixed results. It has been seen on wheat bread that sourdough addition increases bread's hardness [28]. However, it has also been reported that sourdough addition lowers hardness values [29]. Often, high hardness values have been observed in GF products, generally due to the high amount of carbohydrates present in the food matrix [24]. Regarding GF bread, hardness results also often show mixed results. Addition of fermented *Lupinus angustifolius* L. flour (3 g/100 g, 6 g/100 g) increased bread's hardness by 34% and 17%, respectively [30]. Fermented cassava flour has been correlated to increased bread hardness [31]. Hardness results obtained can be observed in Figure 5. All biscuits showed similar results, except for 2 h and 6 h, both being statistically different from each other. Surface response methodology has been used to identify trends in physical and rheological properties of breads. Bread hardness indicated that there is an optimum fermentation time window in which a maximum or minimum value is obtained [32]. Hardness values for 2 h and 6 h could be explained by a similar behavior, 2 h being the maximum value obtained and 6 h the minimum value. Larger cell size formation during baking would have a significant impact on bread's hardness. As seen in Table 1, carbohydrate content decreased while hardness values were maintained in control biscuit and fermented flour biscuit. This can be explained primarily by the baking temperature and moisture content in the food matrix. It has been reported that these two values have a positive correlation on bread crumb's hardness [33].

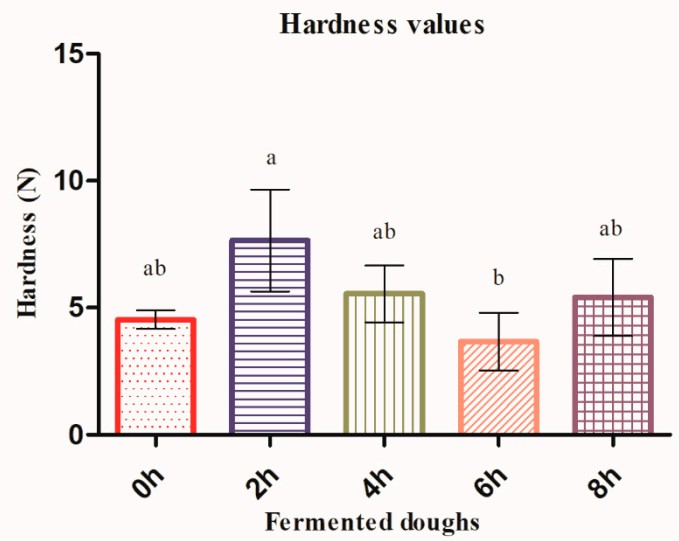

**Figure 5.** Hardness values obtained in biscuits made from fermented flour. Data is mean of three replicates (±standard deviation). **a**, **b** different letters indicate significant differences ($p < 0.05$).

**Table 1.** Results of chemical composition analyzes obtained from biscuits from flours fermented for different amounts of time. Data is mean of three replicates (±standard deviation). a, b different letters indicate significant differences within the same column ($p < 0.05$).

| Fermentation Time | Crude Fiber (%) | Fats (%) | Carbohydrates (%) | Ash (%) | Protein (%) | Moisture (%) |
|---|---|---|---|---|---|---|
| 0 h | 0.40 ± 0.05 [a] | 8.66 ± 1.66 [b] | 56.76 ± 3.16 [a] | 0.08 ± 0.03 [a] | 9.43 ± 0.64 [a] | 24.6 ± 2.31 [a] |
| 2 h | 0.68 ± 0.02 [a] | 14.3 ± 1.08 [a] | 32.9 ± 1.71 [b] | 4.63 ± 0.09 [b] | 12.2 ± 2.08 [a] | 35.1 ± 1.01 [b] |
| 4 h | 0.57 ± 0.05 [a] | 14.2 ± 0.45 [a] | 29.4 ± 7.27 [b] | 4.82 ± 0.14 [b] | 12.2 ± 1.13 [a] | 38.6 ± 6.43 [b] |
| 6 h | 0.40 ± 0.07 [a] | 13.2 ± 0.84 [a] | 35.4 ± 1.10 [b] | 4.79 ± 0.10 [b] | 10.8 ± 0.35 [a] | 35.3 ± 1.15 [b] |
| 8 h | 0.42 ± 0.02 [a] | 12.4 ± 0.16 [a] | 38.0 ± 1.52 [b] | 4.76 ± 0.13 [b] | 10.7 ± 2.16 [a] | 33.5 ± 0.40 [b] |

Both cell number and size changed considerably when comparing to the control biscuit, as can be seen in Figure 6. It has been seen that, with the use of sourdoughs, bread crumb gas cell size increases. However, a transglutaminase (TG) was used to increase fermentation stability [34]. It is

thought that dough acidification could have a direct impact in gas cell formation and size by modifying the interaction between water and other molecules within the food matrix [35]. The decrease of $\overline{G}'$ could be the best explanation for this. It has been observed that the viscous portion of dough mass allows gas cells to expand [36].

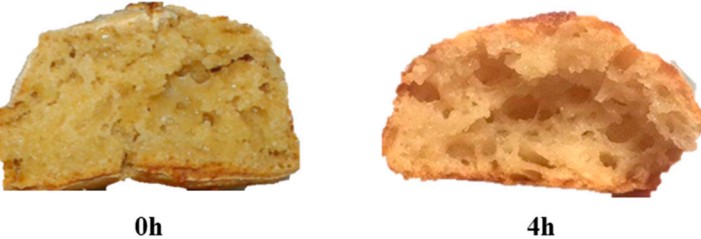

**Figure 6.** Picture of transversal cut of two of the baked biscuits. 0 h control biscuit, 4 h biscuit made from flour fermented for 4 h.

In terms of color, there was no statistical difference in the parameters as can be seen in Table 1. However, there exists a browning effect with a statistical difference between 0 h (blank) and the rest of the biscuits. Contrary results have been obtained where flour formulations were substituted with fermented chestnut flour [6]. L* values ranged from 56.70 to 50.33 increasing along with the chestnut sourdough substitution. However, fermentation times were considerably higher (120 h) than those used in this paper. Similarly, L* values for biscuits made with fermented *Agaricus bisporus* polysaccharides showed a significant difference between their treatments [14]. L* values obtained in this investigation could be explained by the fat content along with maillard reactions [14,32].

*3.4. Chemical Composition Assays*

Results of chemical composition analyses are shown in Table 2. An increase in fat content can be seen by a comparison against 0 h. It has been reported that *L. paracasei* is capable of producing short-chained fatty acids (SCFA). However, levels seen are in the mmol/kg order and in human feces [38]. To the best of our knowledge the highest production of SCFA seen in a sourdough model has peaked at 360 mg/kg [39]. Seeing that there is no statistical difference between treatments, these results suggest that the commercial formulation of the lactic acid bacteria contains an additive. Further studies out of the scope of this work should be carried on determining if there was a production of SCFA.

**Table 2.** Biscuit color analysis results from different fermentation times a, b different letters indicate significant differences within the same column ($p < 0.05$).

| Fermentation Time | Color | | |
|:---:|:---:|:---:|:---:|
| | **L*** | **a*** | **b*** |
| **0 h** | 73.92 ±3.88 [a] | 13.78 ±1.41 [a] | 33.14 ±1.43 [a] |
| **2 h** | 54.99 ±3.17 [b] | 18.39 ±1.71 [a] | 30.92 ±1.73 [a] |
| **4 h** | 54.88 ±2.16 [b] | 20.12 ±1.51 [a] | 32.65 ±1.12 [a] |
| **6 h** | 51.71 ±3.28 [b] | 19.20 ±2.65 [a] | 30.96 ±2.73 [a] |
| **8 h** | 52.28 ±8.64 [b] | 19.69 ±4.63 [a] | 30.66 ±0.54 [a] |

In both a* and b* there was no significant difference between 0 h and treatment biscuits. These results suggest that in order to see significant changes in color, longer fermentation times should be employed as in other works [6,14,37].

Ash content could be explained in a similar manner. Following L * results from Table 1, maillard reactions decreased biscuit's lightness which could explain the difference between ash content in control biscuit (0 h) and those made from fermented flours.

Noticeably, there is no statistical difference between treatments in protein content. This differs from the scientific literature, where it has been seen that flour fermentation with lactic acid bacteria has an effect on both amino acid and protein content [40]. It has been seen that fermentation can significantly increase crude protein content and be detected by the assays done in this paper [41]. However, fermentations last considerably longer in research (72 h).

## 4. Conclusions

Fermentation of cassava flour with *L. paracasei* had an effect on moisture and fat content. Results obtained showed significant differences in chemical properties, which had an impact on the measured rheology properties leading to increased cell size. Hardness values suggest that there is an optimal fermentation time for this property, as previously reported. These results suggest that short fermentation times with *L. paracasei* could be used to improve the rheological characteristics of cassava GF doughs leading to better physical attributes. Fermentation for just 4 h in cassava flour with *L. paracasei*, was enough to show important effects in gas cell size. Nevertheless, longer fermentation times have a significant impact on the nutritional quality of the doughs and biscuits, which should be considered when developing sourdough biscuits seeking enhanced health benefits.

**Author Contributions:** Conceptualization, S.L. and R.B.; methodology, S.L.; validation, J.C., M.F.; formal analysis, S.L.; investigation, S.L.; resources, M.C., R.B.; writing—Original draft preparation, S.L.; writing—Review and editing, R.B., M.C., M.F.; visualization, J.C.; supervision, R.B.; project administration, R.B.; funding acquisition, S.L., R.B. All authors have read and agreed to the published version of the manuscript.

**Funding:** This research was funded by the Mexican Consejo Nacional de Ciencia y Tecnología through a graduate studies scholarship.

**Acknowledgments:** S.L.-G. would like to thank every mentor and colleague at Universidad Autónoma de Coahuila.

**Conflicts of Interest:** The authors declare no conflicts of interest.

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
