# Peer review of "Effect of Short Fermentation Times with Lactobacillus paracasei in Rheological, Physical and Chemical Composition Parameters in Cassava Dough and Biscuits"

_applsci, doi:10.3390/app10041383_

Round 1

Reviewer 1 Report

Abstract

L19-20 “as well as the effects it has in physical and rheology parameters in” change in “and its effects on physical and rheological parameters”

L22 “rheology, physical and chemical composition parameters in both” change in “rheological, physical and chemicals properties in”

L24-25 “Fermentation with L. paracasei had no significant effect in biscuit’s hardness or chemical composition, however gas cell sizes were increased notably” change on biscuit’s hardness and chemical…

Introduction

L32-33 the sentence is not clear, rephrase

L40 change into to develop GF products

L43-45 the sentence is redundant, simplify… “Cassava is one of the most economical source of carbohydrates and energy and its production is increasing”

L50-51 “one of the downsize sourdough present is…” change in “one of sourdough’s downsize is…”

L52 commonly not in italic

L53 on not in

L54 due to the fact that it has been employed in other studies as sourdough starter this sentence is redundant and non correct, you reported studies that use L. paracasei as sourdough starter, but cited only one paper, I suggest to reformulate the sentence: and for its use as sourdough starter…

L55-56 “rheology, physical and chemical composition parameters” change in rheological, physical and chemical properties on dough and biscuits fermented with L. paracasei for short fermentation times

L57 not while, change with and

Materials and Methods

L68 108

L69-70 during 5 min per manufacturers instructions. Five different fermentation times were selected, 0, 2, 4, 6 and 8 h. change in for 5 min, following manufacturers’ instructions. Five different fermentation times were selected (0,2,4,6 and 8h).

L76-78 move to previous paragraph (L71) and eliminate the sentence from Dough to 2.3

L73-74 eliminate from during to above

L80 not during, change with for

L81 change ; with ,

Results and discussion

L115-116 Lactobacillus paracasei in italic (check all the paper…)

L121-123 This could be a potential advantage in industry when looking to reduce processing times while providing the health benefits associated to fermented flours. this sentence isn’t clear check and reformulate…

L124 y??? change this with our

L125 “in the case of fermented flour and 4.92±0.00 in the case of control flour” change: and 4.92±0.00, respectively for fermented and control dough

L125-127 “direct comparisons are hard to make in this kind of results since the elements employed such as flour source and the probiotic microorganisms are different as well as the CFU per g of flour” change: direct comparisons are difficult, because the elements employed, such as flour source, probiotic microrganisms and its concentration, are different

L163 studies…. you cited only one paper

L173 studies…

L181-186 discuss better this data please

L189-190 Did you compare only dough control at t0? I think will be interesting tho check the behaviour of control dough using the same fermentation times of probiotic dough…

L194 in change in of

L197-198 Why not compare the control (not inoculated) with probiotic dough at several fermentation times???

Conclusions

L231 on

L232-236 reformulate this sentence, results aren’t statistically significative???

Reviewer 2 Report

The manuscript studies the effects of lactic acid fermentation on the physicochemical-rheological properties and chemical composition of dough and biscuits from cassava flour. A clear reduction in the loss modulus and storage modulus are observed with fermentation time and attributed to the released lactic acid facilitating interactions of the dough with water molecules. The changes in chemical composition were attributed to the components contained in the probiotics (ash, fatty acids etc.) but not to metabolic products of the fermentation. The results regarding biscuit hardness are contradictory. the manuscript is well written and the experimental design and study are logical. Some of the results must be further explained, before it can be published.

Please correct line 124 "these values are such as 2.90 y 3.5 [16,17]."

In figure 5 there is shown an increase in the hardness of the biscuits at two hours and a decrease at 6 hours, while for all other times it is rather stable. However, no sufficient explanation is provided for this strange behavior. Please explain these fluctuations and in general describe the results in Fig. 5 more thoroughly, as currently they do not make any sense.

Round 2

Reviewer 1 Report

Data reported in this paper are now more clear. Further change aren't required. Check the bibliography for Italics.